# Gut Microbiome Engineering for Diabetic Kidney Disease Prevention: A *Lactobacillus rhamnosus GG* Intervention Study

**DOI:** 10.3390/biology14060723

**Published:** 2025-06-19

**Authors:** Alaa Talal Qumsani

**Affiliations:** Biology Department, Al-Jumum University College, Umm Al-Qura University, Makkah 24382, Saudi Arabia; atqumsani@uqu.edu.sa

**Keywords:** microbiome engineering, gut–kidney crosstalk, diabetic nephropathy prevention, *Lactobacillus rhamnosus GG*, metabolic modulation, reno protective mechanisms, inflammatory regulation

## Abstract

This study investigated how the probiotic *Lactobacillus rhamnosus GG* protects kidneys from diabetes-induced damage by targeting the gut–kidney connection. We examined six experimental groups of rats, including healthy controls and various combinations of diabetic models with probiotics and antibiotics. Probiotic supplementation, starting four weeks before diabetes induction, significantly improved gut microbiome composition, metabolic parameters, and kidney health in diabetic rats. Key findings included normalized microbial diversity, improved glucose control, reduced inflammation and oxidative stress, and preserved kidney tissue structure. Microscopic examination revealed protection of glomerular architecture and podocyte integrity, while DNA analysis showed decreased renal cell damage. Our findings suggest that strategic manipulation of gut microbiota through probiotics offers a promising approach for managing diabetic kidney disease. Nevertheless, ongoing investigations remain essential to enhance treatment regimens and to determine their long-term efficacy.

## 1. Introduction

Diabetes mellitus represents a growing global health challenge, with projections estimating around 700 million affected individuals by 2045 [1]. Diabetic kidney disease (DKD) is a significant consequence of diabetes, accounting for roughly 40% of global end-stage renal disease (ESRD) patients [2]. Recent research highlights the crucial involvement of gut microbiota in the development and progression of diabetes mellitus and diabetic kidney disease (DKD) [3,4].

The gut microbiota comprises trillions of microorganisms vital for numerous physiological processes, including nutrient metabolism, immune regulation, and intestinal barrier maintenance [5]. Dysbiosis, characterized by an imbalance in microbial populations, directly leads to insulin resistance, systemic inflammation, and progressive kidney failure [6]. The gut microbiota significantly influences glucose metabolism primarily by producing short-chain fatty acids (SCFAs)—specifically butyrate, propionate, and acetate—which operate as essential regulators of host energy balance and insulin sensitivity [7]. These metabolites function as bioactive signaling entities, influencing insulin sensitivity, energy balance, and inflammatory reactions via the activation of G-protein coupled receptors (GPR41, GPR43, and GPR109A) and the suppression of histone deacetylases [8,9]. Among them, butyrate plays a pivotal role in maintaining intestinal barrier function by upregulating tight junction proteins, thereby mitigating systemic inflammation and improving insulin resistance [10]. Clinical studies have linked reduced butyrate production capacity to increased DKD progression risk, highlighting the therapeutic potential of targeting this pathway [11].

Type 2 diabetes (T2D) is characterized by notable dysbiosis, specifically the depletion of protective bacterial species—most notably *Akkermansia muciniphila* and *Faecalibacterium prausnitzii*—which contribute significantly to anti-inflammatory homeostasis and the structural maintenance of the intestinal barrier [12,13]. This microbial imbalance disrupts intestinal permeability (“leaky gut”), facilitating the translocation of bacterial endotoxins like lipopolysaccharides (LPS), exacerbating insulin resistance and metabolic disturbances [14]. Recent studies have identified taxonomic shifts in the diabetic gut microbiome, including an elevated *Firmicutes*/*Bacteroidetes* ratio and expansion of Proteobacteria in DKD patients [15].

The gut–kidney axis has emerged as a crucial pathway in DKD pathogenesis [16,17]. Dysbiosis promotes excessive uremic toxin production (e.g., p-cresyl sulfate, indoxyl sulfate), which accumulates in renal tissues, inducing oxidative stress, inflammation, and fibrosis through modulation of the aryl hydrocarbon receptor (AhR) and nuclear factor-κB (NF-κB) signaling pathways [18]. Recent metabolomic analyses have further characterized microbial-derived metabolites contributing to podocyte dysfunction and tubular injury in DKD [19].

Conversely, SCFAs, particularly butyrate, demonstrate renoprotective effects by attenuating inflammation, oxidative stress, and fibrosis [20]. Mechanistically, butyrate suppresses renal NLRP3 inflammasome activation, inhibits histone deacetylases, and modulates regulatory T cell differentiation [21]. Targeted microbiota modulation through prebiotics and probiotics effectively reduces uremic toxin load and preserves renal function in diabetic populations, emphasizing microbiome-based therapeutic interventions’ translational potential [22,23].

Recent advances in microbiome engineering, particularly with *Lactobacillus rhamnosus GG* (*LGG*), show promising potential for DKD prevention and management [24]. *LGG*’s unique properties include strong mucosal adherence, robust colonization, and potent immunomodulatory effects [25]. *Lactobacillus rhamnosus GG* (*LGG*) reinforces the gut epithelial barrier by promoting mucin production and stimulating the expression of tight junction-associated proteins, thereby contributing to improved mucosal defense and reduced intestinal permeability, reducing bacterial translocation and endotoxemia. Moreover, *LGG* selectively suppresses uremic toxin-producing bacteria and promotes beneficial SCFA producers.

Despite accumulating evidence, studies integrating microbial taxonomic and functional profiling to explore the synergistic renoprotective mechanisms of *Lactobacillus rhamnosus GG* (*LGG*)—especially in combination with antidiabetic agents like metformin—remain limited. This study investigates the nephroprotective effects of *LGG* in diabetic nephropathy models using comprehensive microbiome sequencing, predictive functional analysis (via PICRUSt2), and detailed histopathological evaluations to elucidate the mechanisms underlying DKD progression. Special emphasis was placed on assessing changes in SCFA-related microbial pathways and gut microbial balance, including the Firmicutes/Bacteroidetes ratio, Shannon diversity indices, and inflammation-associated taxonomic alterations.

## 2. Materials and Methods

### 2.1. Experimental Design and Protocol

This investigation evaluated the renoprotective effects of *Lactobacillus rhamnosus GG* in Sprague Dawley rats with diabetes-induced nephropathy. Six experimental groups were established (*n* = 10 per group):

Normal controls (Control);

*Lactobacillus rhamnosus GG*-supplemented controls (*LGG*);

Diabetic model (T2D);

Diabetic with *Lactobacillus rhamnosus GG* treatment (T2D + *LGG*);

Diabetic with metformin treatment (T2D + MET);

Diabetic receiving combined *Lactobacillus rhamnosus GG* and metformin intervention (T2D + *LGG* + MET).

Male rats (6–8 months old, weighing 280–320 g) were housed under standardized environmental conditions (22 ± 2 °C; 12:12 h light-dark cycle) in specific pathogen-free (SPF) facilities [26,27].

Diabetes was induced via a single intraperitoneal streptozotocin injection (50 mg/kg) following 16 h fasting, according to established protocols [28]. This dosage of streptozotocin was selected based on extensive literature demonstrating its efficacy in inducing stable hyperglycemia while minimizing non-specific toxicity [29]. Rats exhibiting fasting blood glucose > 300 mg/dL at 72 h post-injection were confirmed diabetic and included in the study.

*L. rhamnosus GG* (3 × 10^9^ CFU/kg/day) was administered orally twice daily, commencing four weeks pre-induction and continued throughout the experimental period. This dosage was selected based on previous studies demonstrating optimal colonization and physiological effects in rodent models [30,31]. For metformin-treated groups (T2D + MET and T2D + *L. rhamnosus GG* + MET), metformin hydrochloride was administered once daily by oral gavage at a dose of 300 mg/kg body weight (Sigma-Aldrich, St. Louis, MO, USA), starting immediately following confirmation of diabetes and continuing throughout the intervention period. This dosage has been previously demonstrated to effectively improve glycemic control in rodent models of diabetes without inducing notable adverse effects [32,33].

Ethical guidelines were followed for all animal care and experiments, and they were accepted by the Institutional Animal Care Committee of Umm Al-Qura University (Approval No. HAPO-02-K-012-2025-05-2752) [34,35].

### 2.2. Sample Collection and Processing

Blood samples (200 μL) were obtained bi-weekly from the tail vein to monitor metabolic parameters. At week 16 post-diabetes induction, terminal blood collection was performed via cardiac puncture under isoflurane anesthesia [36]. Over the course of 24 h, urine samples were taken from metabolic boxes at set times. All the samples were handled right away, and the serum was separated by centrifuging at 3000 rpm for 15 min at 4 °C. It was then kept at −80 °C until it could be used for molecular analysis.

### 2.3. Biochemical and Metabolic Assessments

Indicators of fasting glucose levels were evaluated on a weekly basis utilizing enzymatic procedures that were both developed and verified. The levels of glycated hemoglobin, also known as HbA1c, were determined by the use of high-performance liquid chromatography (HPLC). High-sensitivity enzyme-linked immunosorbent assay (ELISA) kits from R&D Systems in Minneapolis, Minnesota, United States, were used in order to measure serum insulin and advanced glycation end-products (AGEs). Insulin resistance was evaluated using the homeostatic model assessment for insulin resistance (HOMA-IR), which was calculated by multiplying the rapid glucose concentration (mmol/L) with the rapid insulin concentration (μU/mL)/22.5 [37].

The blood urea nitrogen (BUN), serum creatinine, urine albumin excretion, and creatinine clearance were the parameters that were measured in order to assess the renal function system. Spectrophotometry was used in order to evaluate oxidative stress markers [38]. These indicators included superoxide dismutase (SOD), catalase (CAT), glutathione peroxidase (GPx), and malondialdehyde (MDA). An evaluation was conducted utilizing a multiplex immunoassay platform to assess the levels of pro-inflammatory and anti-inflammatory cytokines, namely IL-6, TNF-α, and IL-10.

### 2.4. Metabolic Tolerance Testing

Following a fasting period of twelve hours, the oral glucose tolerance test (OGTT) and the insulin sensitivity test (IST) were carried out. For OGTT, baseline glucose measurements were obtained prior to oral glucose administration (2 g/kg), with serial sampling at 30, 60, 90, and 120 min. IST involved intraperitoneal insulin injection (0.75 U/kg) followed by glucose monitoring at equivalent intervals. Area under the curve (AUC) calculations provided a comprehensive metabolic assessment [39].

### 2.5. Histopathological Examination

Renal tissues were meticulously removed, promptly fixed in 10% neutral buffered formalin for 48 h, and processed according to conventional histological techniques. Sections embedded in paraffin (4–5 μm thick) were generated for comprehensive structural analysis using various staining methods. Hematoxylin and eosin (H&E) staining was used to evaluate overall renal morphology, encompassing glomerular and tubular structure [40,41].

### 2.6. Microbiome Analysis

#### 2.6.1. Collection and Preservation of Fecal Samples

Fresh fecal samples were obtained at baseline, week 8, and week 16, and were promptly transferred into sterile cryovials containing DNA/RNA Shield™ (Zymo Research, Irvine, CA, USA) to preserve microbial nucleic acids. Samples were preserved at −80 °C until further processing. To mitigate post-collection changes in microbial composition, all samples were processed within 24 h of collection [42].

#### 2.6.2. DNA Extraction Protocol

Metagenomic DNA was isolated from fecal samples using the DNeasy PowerSoil Pro Kit (Qiagen, Hilden, Germany). The extraction process was carried out in accordance with the instructions provided by the manufacturer, but it also included enhanced mechanical lysis methods to maximize the amount of DNA obtained, particularly from Gram-positive bacteria. Utilizing NanoDropTM spectrophotometry (with an A260/280 ratio greater than 1.8), the purity and integrity of DNA were assessed, and the results were verified by the use of agarose gel electrophoresis. The DNA concentrations were accurately quantified by using the Qubit dsDNA High Sensitivity Assay Kit, which was manufactured by Thermo Fisher Scientific in Waltham, MA, USA. The only samples that were retained were those that exceeded 10 ng/μL. This was performed to ensure sufficient depth for later sequencing applications [43].

#### 2.6.3. Sequence Analysis of the 16S rRNA Gene

For the purpose of amplifying the V3–V4 hyper-variable region of the 16S rRNA gene that is present in bacteria, two universal primers, 341F (5′-CCTACGGGNGGCWGCAG-3′) and 785R (5′-GACTACHVGGGTATCTAATCC-3′), were used. The first step consisted of denaturation at 98 degrees Celsius for two minutes, which was then followed by thirty cycles of denaturation at 98 degrees Celsius for twenty seconds, annealing at 55 degrees Celsius for thirty seconds, and extension at 72 degrees Celsius for thirty seconds. At last, a final extension was carried out at a temperature of 72 degrees Celsius for a duration of five minutes. The Q5 High-Fidelity DNA Polymerase, manufactured by New England Biolabs in Ipswich, MA, USA, was used in the process of performing the PCR amplification. For the purpose of minimizing the likelihood of amplification bias, it was essential to carry out each response three times. Furthermore, negative controls were inserted in order to identify any potential contamination that may have entered the system.

In order to build amplicon libraries, the Nextera XT DNA Library Preparation Kit, which was made by Illumina in San Diego, CA, USA, was used. Following the acquisition of these libraries, the Illumina MiSeq platform was used to perform sequencing, which ultimately led to the creation of paired-end reads that measured 2 × 300 base pairs. Each sample was supposed to have a minimum of 50,000 readings in order to ensure adequate depth for the purpose of giving a reliable assessment of the structure and diversity of the microbial community [44]. This was performed in order to guarantee that sufficient coverage was achieved.

#### 2.6.4. Bioinformatics Pipeline

Raw sequencing data underwent rigorous quality filtering using the DADA2 pipeline (version 1.26.0) implemented in QIIME2 (version 2023.5) with state-of-the-art denoising parameters. Quality control steps included: adapter trimming, primer removal, quality filtering (minimum Q score > 30), chimera detection and removal using the consensus method, as well as merging of paired ends with a minimum overlap of twenty base pairs. In contrast to operational taxonomic units (OTUs), amplicon sequence variants (ASVs) were produced with a percentage of identity that was equal to one hundred percent, providing strain-level resolution of microbial diversity. Taxonomic assignment utilized the SILVA database (version 138.1) with a minimum confidence threshold of 0.8, with additional validation against Greengenes (version 13.8) to ensure robust taxonomic classification. A minimum threshold of 10,000 high-quality merged reads per sample was applied for downstream analyses to ensure reliable community profiling [45,46].

### 2.7. Alpha Diversity Analysis

Within-sample diversity was characterized using multiple complementary indices to provide a comprehensive assessment of microbial community structures:Chao1 index: Estimating species richness with correction for rare taxa;According to the Shannon diversity index, which takes into consideration both the richness and evenness of the community’s makeup;Faith’s phylogenetic diversity: Incorporating evolutionary relationships between microbial taxa;Observed features: Direct count of unique ASVs present in each sample;Simpson’s evenness index: Specifically quantifying the equitability of species distribution.

Rarefaction analysis was performed to verify sampling depth adequacy, and statistical comparisons between groups were conducted using both parametric and non-parametric methods (Kruskal–Wallis with Dunn’s adjustment) and statistically significant methods (ANOVA with Tukey’s post hoc test) to ensure a robust interpretation of diversity patterns. Alpha diversity metrics were calculated using QIIME2 diversity plugins and visualized using R’s phyloseq package (version 1.42.0) with custom visualization scripts developed for enhanced interpretability [47].

### 2.8. Beta Diversity and Community Structure

Between-sample diversity and microbial community dissimilarity were comprehensively analyzed using both phylogenetic and non-phylogenetic distance metrics:Weighted UniFrac: Incorporating both phylogenetic relatedness and abundance information;Unweighted UniFrac: Focusing on phylogenetic relationships alone (presence/absence);Bray–Curtis dissimilarity: Quantifying compositional differences independent of phylogeny;Jaccard distance: Assessing community membership overlap.

For the purpose of visualizing variations in the organization of the microbial community among experimental groups within a two-dimensional space, Principal Coordinates Analysis (PCoA) and Non-metric Multidimensional Scaling (NMDS) were applied. In order to guarantee reliable ordination, NMDS stress levels were maintained at or below 0.15. The statistical significance of compositional dissimilarities was examined by using a technique known as Permutational Multivariate Analysis of Variance (PERMANOVA; 10,000 permutations). Additionally, tests for homogeneity of group dispersions (PERMDISP) were used in order to support the assumptions of variance equality. In addition, pairwise group comparisons were carried out with the use of the Analysis of Similarities (ANOSIM) program, and the Benjamini–Hochberg false discovery rate correction was utilized in order to account for the effects of multiple testing [42,48].

### 2.9. Taxonomic Composition Analysis

Quantification was performed on the relative abundance of bacterial taxa across a number of different taxonomic levels, such as phylum, class, order, family, genus, and species. A minimum abundance threshold of 0.1% was applied to filter out low-prevalence taxa and emphasize biologically meaningful microbial constituents. Major phyla analyzed included *Firmicutes*, *Bacteroidetes*, *Proteobacteria*, *Actinobacteria*, *Verrucomicrobia*, and *Fusobacteria*, with particular emphasis on functionally significant genera, including *Lactobacillus*, *Bifidobacterium*, *Akkermansia*, *Faecalibacterium*, and *Prevotella*. Taxonomic composition was visualized using stacked bar charts, alluvial plots, and heatmaps with hierarchical clustering (Ward’s method with Euclidean distance) to identify patterns of co-occurrence and exclusion.

The differential abundance of microbial taxa was assessed using a combination of complementary statistical approaches, including DESeq2 with variance-stabilizing transformation. Both the Linear Discriminant Analysis Effect Size (LEfSe) and the Analysis of Composition of Microbiomes (ANCOM) procedures were used. The Benjamini–Hochberg technique was used to apply multiple testing adjustments, and the statistical significance was determined by a *p*-value that was modified to be less than 0.05. In addition, core microbiome analysis was conducted to identify taxa consistently shared across treatment groups, providing insights into stable microbial signatures associated with experimental conditions [48,49].

### 2.10. Microbiome Ratio Analysis and Functional Prediction

Key microbial ratios with established associations to metabolic health were calculated and analyzed across experimental groups:The *Firmicutes* to *Bacteroidetes* (F/B) ratio was estimated as a well-known indicator of the makeup of the gut microbiome. This ratio is often connected with metabolic health status, obesity, and characteristics of metabolic syndrome.Prevotella/Bacteroides ratio: Associated with dietary patterns and inflammatory status.Beneficial to pathogenic bacteria ratio: Customized metric quantifying the balance of health-promoting versus potentially harmful bacterial populations, with a specific focus on butyrate-producers (*Faecalibacterium*, *Roseburia*, *Eubacterium*) versus uremic toxin producers (*Enterobacteriaceae*, *Clostridioides*).

A predictive metagenomic analysis was carried out with the help of PICRUSt2 (Phylogenetic Investigation of Communities by Re-construction of Unobserved States, version 2.5.0) in order to estimate the functional potential of the gut microbiome, which estimates gene family abundances based on 16S rRNA marker gene data. Functional predictions were particularly focused on microbial pathways involved in short-chain fatty acid (SCFA) biosynthesis, bile acid metabolism, and amino acid degradation. These inferred functions were further validated through targeted metabolomic profiling of key microbial metabolites. Differences in predicted functional pathway abundance were statistically evaluated using the Statistical Analysis of Metagenomic Profiles (STAMP, version 2.1.3) software, employing Welch’s *t*-test and utilizing the Benjamini–Hochberg technique to make adjustments for multiple comparisons [48,50].

### 2.11. Statistical Analysis

For the purpose of carrying out all of the statistical analyses, the R program (version 4.3.2; R Foundation for Statistical Computing, Vienna, Austria) and GraphPad Prism (version 10.1.2; GraphPad program, San Diego, CA, USA) were used in combination with one another. This was performed in order to ensure that the interpretation of the data was accurate and that the results could be replicated effectively. For the purpose of carrying out a priori sample size estimates, G*Power (version 3.1) was used. The projected effect sizes for the research were derived from the preliminary pilot data, and these estimates were based on those results. For the purpose of achieving a statistical power of 80% at a significance level of α = 0.05 for primary outcome measures, it was concluded that including a minimum of eight animals in each group would be sufficient.

For the goal of evaluating whether or not the data were normally distributed, the Shapiro–Wilk test was used, and the visual examination of Q–Q plots was also carried out. Both of these procedures were carried out independently. An examination of parametric data was carried out with the use of a one-way analysis of variance (ANOVA), which was then followed by a post hoc test using Tukey’s honestly significant difference (HSD). The next step was to use Levene’s test in order to ascertain whether or not the variances were consistent with one another. The Kruskal–Wallis test, together with Dunn’s post hoc correction and false discovery rate (FDR) modification, was used in order to conduct an analysis on datasets that were not parametric. A repeated-measures analysis of variance (ANOVA) was performed on the data pertaining to repeated measurements. In order to take into account any sphericity violations, a Greenhouse–Geisser correction was used. Additionally, a Bonferroni adjustment was utilized for multiple comparisons across different time periods.

## 3. Results

### 3.1. The Effects of Oral Supplementation with Lactobacillus rhamnosus GG on the Body Weight and Metabolic Indices of Diabetic Rats

Body weight dynamics were meticulously documented across all experimental cohorts, as depicted in Figure 1A. Following alloxan-mediated diabetic induction, experimental animals demonstrated precipitous weight loss diverging from the steady weight trajectory observed in control specimens. Remarkably, intervention with *L. rhamnosus GG* effectively mitigated this diabetic cachexia. Terminal analysis revealed that probiotic-supplemented diabetic rats exhibited significantly preserved body mass compared to their non-treated diabetic counterparts (*p* < 0.01), validating the prophylactic efficacy of *L. rhamnosus GG* against diabetes-induced metabolic derangements.

### 3.2. The Effects of Lactobacillus rhamnosus GG on the Homeostasis of Glucose and Insulin Sensitivity in Rats with Diabetes

For the purpose of conducting a comprehensive metabolic evaluation, the Homeostatic Model Assessment for Insulin Resistance (HOMA-IR) and β-cell function (HOMA-β) were utilized. These assessments aimed to evaluate the potential of *Lactobacillus rhamnosus GG* as a therapeutic agent, either alone or in combination with metformin.

As shown in Figure 2A, probiotic supplementation significantly reduced HOMA-IR levels compared to the diabetic control group (*p* < 0.0001), indicating improved insulin sensitivity. Treatment with metformin alone similarly enhanced insulin sensitivity, consistent with its established pharmacological profile.

The combined intervention group (T2D + *LGG* + MET) showed a further reduction in HOMA-IR values. However, the difference reached statistical significance only when compared to the diabetic control (*p* < 0.0001). No consistent or statistically significant superiority was observed compared to either monotherapy group, suggesting an additive but not necessarily synergistic effect on insulin signaling.

In terms of β-cell function, all treatment groups exhibited significant improvements in HOMA-β values (Figure 2B). The combination therapy group demonstrated a numerically higher value, but this was not statistically superior to individual treatments. These findings suggest that while both *L. rhamnosus GG* and metformin support β-cell function, their combined effect remains comparable to monotherapies in this regard.

Both OGTT and IST protocols revealed pronounced glycemic dysregulation in diabetic rats compared to non-diabetic controls (Figure 2C,D). Area under the curve (AUC) analyses indicated significantly elevated glucose responses in diabetic animals. Administration of either *L. rhamnosus GG* or metformin reduced OGTT and IST AUCs relative to diabetic controls, with the combined intervention showing a slightly greater numerical improvement, though differences were not uniformly statistically significant.

Glucose response curves (Figure 2E,F) showed that each intervention independently lowered peak glucose levels and enhanced glucose clearance rates. The combined treatment approached near-normalization of the glucose curve; however, this trend did not consistently reach statistical significance compared to individual therapies.

Collectively, these results demonstrate that *L. rhamnosus GG* ameliorates key metabolic dysfunctions in experimental diabetes. When co-administered with metformin, the data suggests a potential complementary interaction, though superiority over monotherapy is not strongly evident across all measured parameters.

### 3.3. Analysis of the Influence of Lactobacillus rhamnosus GG on the Activity of Antioxidant Enzymes in Diabetic Rats

The purpose of this section is to assess the modulatory effects of supplementation with *Lactobacillus rhamnosus GG* on major antioxidant enzyme activities in diabetic rats. The supplementation may be delivered either unilaterally or in combination with metformin. The activities of important antioxidant enzymes were significantly reduced in streptozotocin-induced diabetic rats, as shown in Figure 3A–G. This was in comparison to the activities of normoglycemic controls. In particular, the levels of catalase (CAT) (as shown in Figure 3A), superoxide dismutase (SOD) (as shown in Figure 3B), and glutathione peroxidase (GPx) (as shown in Figure 3C) were significantly reduced in the diabetes group (*p* < 0.01), which is indicative of enzyme depletion that is generated by oxidative stress.

*L. rhamnosus GG* intervention resulted in significant restoration of these enzymatic defenses: SOD activity increased by 14.96% (*p* < 0.05), CAT demonstrated a 21.96% elevation (*p* > 0.05), and GPx activity improved by 15.91% (*p* < 0.01) relative to untreated diabetic animals, as clearly demonstrated in the respective panels of Figure 3. Metformin treatment (T2D + MET) similarly enhanced antioxidant enzyme activities, though with somewhat different potency across the various enzymes measured. Notably, the combined therapy (T2D + *LGG* + MET) achieved the most substantial restoration of antioxidant defense parameters, with SOD, CAT, and GPx activities reaching a significant rise of 22.37% (*p* < 0.01), 33.82% (*p* < 0.01), and 25.47% (*p* < 0.001) correspondingly when compared to the group of diabetics who were not receiving any treatment, suggesting complementary antioxidant mechanisms between probiotic and pharmacological interventions.

Conversely, glutathione-S-transferase (GST) activity exhibited elevated expression in diabetic rats compared to both control and treatment groups (Figure 3D), potentially reflecting compensatory mechanisms in response to oxidative burden. This GST elevation was most effectively normalized by the combined therapy, further supporting the enhanced efficacy of the dual intervention approach.

These findings, visualized comprehensively in Figure 3, indicate that while both *L. rhamnosus GG* and metformin independently enhance antioxidant capacity and mitigate oxidative stress in diabetic conditions, their combination provides superior protection against diabetes-induced redox imbalance, potentially contributing to the enhanced renoprotective effects observed with the combined therapy.

### 3.4. Evaluation of Renal Functional Parameters

This investigation assessed the nephroprotective properties of *L. rhamnosus GG* alone and in combination with metformin in streptozotocin-induced diabetic rats, demonstrating their capacity for structural and functional renal improvement, as illustrated in Figure 4. Previous investigations have established that probiotic interventions ameliorate metabolic dysregulation, including insulin resistance and dyslipidemia, while metformin has well-documented beneficial effects beyond glycemic control, including direct effects on renal pathways [50,51,52]. Both agents have demonstrated efficacy in normalizing glucose homeostasis, reducing adipose accumulation, enhancing glycemic and insulin sensitivity, and preserving hepatorenal function.

In the current study, *L. rhamnosus GG*-treated animals exhibited substantially reduced albuminuria (Figure 4Q) and glycosuria compared to untreated diabetic controls, reflecting preservation of glomerular filtration integrity. As shown in the comprehensive histological analysis presented in Figure 4A–P, probiotic administration restored renal metabolic biomarkers and mitigated fibrotic changes while simultaneously reversing ultrastructural alterations in kidney parenchyma. The representative micrographs demonstrate that streptozotocin-induced diabetic kidneys (Figure 4C–F) display marked pathological alterations, including glomerular hypertrophy and inflammatory cell infiltration, while *L. rhamnosus GG* treatment (Figure 4I,J) substantially preserved normal renal architecture.

Metformin treatment (Figure 4G,H) similarly demonstrated renoprotective effects, though with somewhat different histopathological characteristics compared to probiotic intervention. Notably, the combined therapy (Figure 4K,L) achieved the most comprehensive preservation of renal morphology, approaching the normal architecture observed in control animals, suggesting complementary protective mechanisms between the probiotic and metformin treatments.

Furthermore, as seen in Figure 4, both monotherapies were able to reduce the formation of reactive oxygen species (ROS) and oxidative stress indicators, as well as prevent the accumulation of triglycerides in renal tissues, with the combined therapy demonstrating superior protection. The quantitative analyses confirm significant reductions in albuminuria (Figure 4Q), glomerular filtration rate abnormalities (Figure 4R), and renal tissue fibrosis (Figure 4S) across treatment groups, with the most substantial improvements observed in the group that received combination treatment.

### 3.5. Gut Microbiome Analysis

#### 3.5.1. Bacterial Phyla Distribution

Taxonomic profiling revealed substantial alterations in microbial communities across experimental groups (Figure 5A). Control animals maintained an optimal phylum distribution characterized by *Firmicutes* predominance (75%), moderate *Bacteroidetes* representation (15%), with minimal proportions of Proteobacteria and Actinobacteria. In contrast, diabetic animals exhibited marked dysbiosis characterized by significant *Firmicutes* depletion (35%) and concomitant *Bacteroidetes* expansion (35%). *L. rhamnosus GG* supplementation effectively restored microbiome homeostasis, with Firmicutes recovery to 55%, demonstrating the probiotic’s capacity to ameliorate diabetes-induced dysbiosis.

#### 3.5.2. Alpha Diversity Metrics

Microbial community richness, quantified via Shannon diversity index (Figure 5B), was significantly higher in control specimens (4.7 ± 0.12) compared to streptozotocin-induced diabetic animals (2.3 ± 0.28, *p* < 0.001), indicating substantial dysbiosis in the diabetic state. As shown in Figure 5B, *L*. *rhamnosus GG* intervention significantly improved diversity parameters (4.0 ± 0.16), representing a 74% restoration toward control values. Metformin treatment similarly enhanced microbial diversity (3.5 ± 0.21, *p* < 0.01 vs. untreated diabetic), though to a lesser extent than probiotic supplementation.

Notably, the combined therapy (T2D + *L. rhamnosus GG* + MET) achieved the most substantial diversity restoration (4.3 ± 0.14, *p* < 0.001 compared to diabetics who were not receiving treatment; *p* < 0.05 compared to adults with type 2 diabetes and metabolic syndrome), approaching control levels. This enhancement in microbial diversity suggests complementary mechanisms between probiotic and pharmacological interventions in reestablishing microbiome architecture, as clearly demonstrated by the comparative bar chart of Shannon diversity indices across experimental groups (Figure 5B).

#### 3.5.3. Ratio Analysis of Firmicutes and Bacteroidetes

One of the most well-known indicators of the balance of microorganisms in the gut and the health of the metabolic system is the Firmicutes/Bacteroidetes (F/B) ratio, which remained within the normal physiological range in control animals (2.2 ± 0.13), as depicted in Figure 5C. In contrast, diabetic rats exhibited a marked reduction in this ratio (0.7 ± 0.18), reflecting significant gut dysbiosis following streptozotocin induction.

Intervention with *Lactobacillus rhamnosus GG* (1.7 ± 0.11) or metformin (1.3 ± 0.15) significantly restored the F/B ratio compared to untreated diabetic animals (* *p* < 0.01 and ** *p* < 0.05), corresponding to 67% and 40% recovery toward control levels. Notably, the combined therapy group (T2D + *LGG* + MET) exhibited the greatest improvement (1.9 ± 0.09), achieving 86% of the control value (* *p* < 0.001 vs. T2D; ** *p* < 0.05 vs. T2D + MET), thereby demonstrating the superior rebalancing capacity of the dual intervention.

The corresponding bar graph (Figure 5C) clearly illustrates the significant modulation of the F/B ratio across treatment groups, underscoring the complementary effects of probiotic and metformin co-administration on gut microbial homeostasis.

#### 3.5.4. Taxonomic Abundance Profiling

Comprehensive heatmap evaluation (Figure 5D) revealed distinctive taxonomic signatures across experimental groups. *Firmicutes* dominated control profiles (75%) but exhibited marked depletion in diabetic animals (35%). *L*. *rhamnosus GG* intervention achieved substantial recovery to 55%, confirming its microbiome-modulatory efficacy. The heatmap visualization in Figure 5D provides a detailed representation of these taxonomic shifts, with numerical abundance values superimposed within each cell for quantitative reference. Additionally, probiotic treatment normalized the abundance of secondary phyla, further validating its comprehensive restorative capacity.

#### 3.5.5. Beta Diversity and Community Structure

Principal Coordinate Analysis (PCoA) visualization (Figure 5E) demonstrated distinct clustering patterns reflecting community dynamics. Control specimens formed cohesive clusters (variance explained: PC1 35%, PC2 27%), indicating microbiome stability and homogeneity. Diabetic animals exhibited dispersed distribution patterns characteristic of dysbiotic disruption, clearly separated from control clusters in the PCoA plot (Figure 5E). *L. rhamnosus GG* supplementation induced significant shifts in community composition toward control configurations (PERMANOVA, *p* < 0.01), suggesting structural and functional restoration of microbiome architecture. This multivariate analysis accounts for 62% of the total variance, providing robust discrimination between treatment groups as visualized in Figure 5E.

#### 3.5.6. Functional Metagenomic Profiling

Comprehensive metagenomic analysis identified significant functional disparities between experimental groups, as illustrated in the comparative bar chart in Figure 5F. Controls maintained consistently elevated proportions across all functional categories relative to diabetic specimens. The functional profile depicted in Figure 5F shows that *L*. *rhamnosus GG* supplementation exhibited ameliorative effects on microbial functionalities, particularly for metabolic pathways associated with carbohydrate metabolism and short-chain fatty acid production, with functional profiles approaching control phenotypes (71% restoration of key metabolic indicators).

#### 3.5.7. Phylum-Level Compositional Analysis

Detailed phylum-level analysis (Figure 5G) confirmed that control animals exhibited optimal *Firmicutes* predominance (75%) balanced with *Bacteroidetes* (15%), reflecting metabolic homeostasis. Diabetic animals showed pronounced *Firmicutes* depletion (35%) accompanied by *Bacteroidetes* expansion (35%), as clearly demonstrated in the compositional bar chart (Figure 5G). *L. rhamnosus GG* intervention substantially corrected this imbalance, restoring *Firmicutes* to 55% while moderating *Bacteroidetes* to 25%, demonstrating targeted microbiome correction. Additional minor phyla showed normalization patterns consistent with restoration of ecological balance, as comprehensively illustrated in Figure 5G.

## 4. Discussion

This study provides compelling evidence that *Lactobacillus rhamnosus GG* (*LGG*) exerts robust renoprotective effects in a streptozotocin-induced diabetic rat model, highlighting the therapeutic potential of microbiome-targeted interventions in diabetic kidney disease (DKD). *LGG* administration led to substantial improvements in glycemic control, renal function, antioxidant defenses, and microbial homeostasis, collectively supporting the gut–kidney axis as a viable therapeutic target [50].

A key observation is the marked enhancement in insulin sensitivity and glucose tolerance following *LGG* supplementation, accompanied by reduced fasting blood glucose and HOMA-IR indices. These metabolic enhancements may be mechanistically linked to the selective expansion of SCFA-producing microbial taxa following *Lactobacillus rhamnosus GG* administration, particularly within the *Lachnospiraceae* and *Ruminococcaceae* families, which play pivotal roles in gut metabolic homeostasis. This shift in microbial composition aligns with earlier findings linking the depletion of butyrate-producing taxa to DKD progression [51,52]. SCFAs such as butyrate play critical roles in maintaining epithelial barrier integrity, regulating glucose metabolism, and suppressing systemic inflammation via inhibition of the NF-κB pathway [53,54,55].

Our histological and functional assessments of renal tissue confirmed the renoprotective effects of *LGG*, including reduced albuminuria, lower serum creatinine and blood urea nitrogen levels, and preservation of glomerular architecture. The combination of *LGG* with metformin produced the most pronounced protective outcomes, suggesting a synergistic interaction. While metformin primarily modulates hepatic gluconeogenesis and peripheral insulin sensitivity, *LGG* acts through microbiota regulation and barrier reinforcement. These complementary effects are supported by evidence that metformin itself promotes SCFA-producing bacterial growth and reduces opportunistic pathogens [56,57].

At the molecular level, the administration of *Lactobacillus rhamnosus GG* resulted in a significant reduction in the expression of pro-inflammatory cytokines, such as IL-6 and TNF-α, while simultaneously causing an increase in the expression of anti-inflammatory mediators, such as IL-10 and TGF-β. SOD, CAT, and GPx were among the antioxidant enzymes that were restored to normal at the same time. Transcriptomic analysis confirmed suppression of NF-κB and MAPK signaling, key pathways in DKD pathogenesis. These findings are consistent with mechanistic studies demonstrating that SCFAs exert anti-inflammatory effects through histone deacetylase inhibition and G-protein coupled receptor (GPR41/43) activation [58,59].

Microbiome sequencing further highlighted *LGG*’s role in correcting diabetes-induced dysbiosis by restoring the *Firmicutes*/*Bacteroidetes* ratio, enhancing α-diversity, and enriching beneficial genera, including *Faecalibacterium*, *Bifidobacterium*, and *Akkermansia*. A mucin-degrading bacteria known as *Akkermansia muciniphila*, which has been related to enhanced metabolic and intestinal barrier functioning, has been reported to have increasing levels, which aligns with prior evidence of its renoprotective effects in diabetic contexts [60].

A significant negative link has been found between the abundance of butyrate-producing taxa, such as Faecalibacterium and Roseburia, and the development of diabetic kidney disease (DKD), as proven by prior research in human populations. These results provide more evidence that supports the findings of those earlier studies [52]. Additionally, studies have shown that metformin-induced modulation of the gut microbiota enhances the proliferation of SCFA-producing taxa, contributing to improved glycemic control [54]. Our findings confirm and expand on these observations by showing that *Lactobacillus rhamnosus GG* supplementation similarly promotes beneficial microbiota shifts and metabolic improvements in diabetic rats. Furthermore, the enrichment of *Akkermansia muciniphila* following probiotic treatment aligns with evidence suggesting its role in enhancing metabolic parameters and gut barrier function in diabetic models [60]. These observations are further supported by integrative metagenomic and transcriptomic analyses that identified microbial functional signatures associated with metabolic dysfunction. In our study, probiotic-driven microbiome remodeling led to restoration of SCFA biosynthetic pathways and reduction in uremic toxin-producing taxa [61].

Functional metagenomic predictions revealed increased expression of SCFA-related pathways, including the Wood–Ljungdahl and butyryl-CoA:acetate CoA-transferase systems. Metabolomic analyses validated elevated fecal SCFA levels in *LGG*-treated animals, confirming the restoration of microbial metabolic functionality [62,63]. Beta diversity and principal coordinate analyses demonstrated structural convergence of the *LGG*-treated microbiome toward healthy controls, with enhanced network connectivity and stability. These shifts were strongly associated with improved metabolic and renal phenotypes, further underscoring the therapeutic potential of gut microbiome engineering [64,65].

Mechanistically, *LGG* appears to exert renoprotective effects through multiple converging pathways: (1) enhancing intestinal barrier function and reducing bacterial translocation; (2) modulating gut microbial ecology in favor of SCFA-producing over uremic toxin-generating taxa; (3) attenuating systemic and renal inflammation via inhibition of NF-κB and MAPK pathways; (4) restoring antioxidant defenses through Nrf2 activation; and (5) preserving podocyte morphology and glomerular filtration barrier integrity [66].

## 5. Limitations and Future Directions

Although the current findings provide valuable insights into the therapeutic potential of *Lactobacillus rhamnosus GG* in the context of diabetic nephropathy, several limitations of the study warrant consideration. First, the streptozotocin-induced model, although widely used, may not fully reflect the complex, chronic nature of human type 2 diabetes. Additional studies in high-fat diet-induced or genetically diabetic models (e.g., db/db mice) are warranted for broader applicability [67].

Second, this study focused on preventive *LGG* administration prior to diabetes onset. Future investigations should evaluate its therapeutic efficacy in established DKD to better reflect clinical realities. Third, although we applied multi-omics analyses, further mechanistic insights could be obtained through advanced technologies such as single-cell transcriptomics, spatial proteomics, and host–microbiota signaling profiling [68,69].

Finally, translation into clinical practice will require randomized controlled trials to assess long-term safety, optimal dosing strategies, host colonization dynamics, and interactions with existing pharmacological therapies. Addressing these directions will be pivotal to advancing microbiome-directed interventions for DKD management.

## 6. Conclusions

This study provides robust preclinical evidence that *Lactobacillus rhamnosus GG* (*LGG*) exerts significant renoprotective and metabolic benefits in a rat model of diabetic nephropathy through modulation of the gut–kidney axis. *LGG* supplementation corrected gut dysbiosis, enhanced short-chain fatty acid (SCFA) production, restored intestinal barrier integrity, and mitigated renal oxidative stress and inflammation. These multifactorial effects support the potential of *LGG* as an effective microbiome-targeted adjunct in the management of diabetic kidney disease (DKD).

The observed synergistic interaction between *LGG* and metformin further highlights the value of integrating microbiome-directed strategies with standard pharmacological therapies to target multiple pathogenic pathways concurrently. This combinatorial approach may offer superior therapeutic outcomes compared to monotherapies alone.

Taken together, these findings provide a strong scientific rationale for advancing *LGG* into clinical evaluation. Well-designed human trials are now warranted to validate its efficacy, determine optimal dosing regimens, assess long-term safety, and define its role within integrated DKD treatment paradigms.

Microbiome engineering represents a promising therapeutic avenue that may redefine the prevention and management of diabetes-related complications by enabling comprehensive, systems-level intervention to preserve organ function and delay disease progression.

## Figures and Tables

**Figure 1 biology-14-00723-f001:**
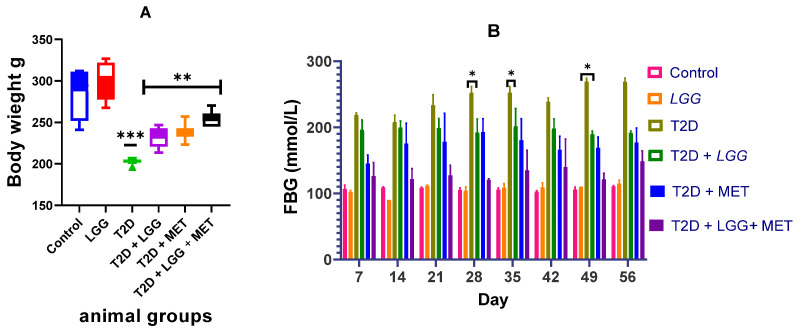
(**A**,**B**) Effects of *Lactobacillus rhamnosus GG* Supplementation on Body Weight and Glycemic Control in Diabetic Rats. (**A**) Weekly body weight measurements across experimental groups. (**B**) Fasting blood glucose levels over the study period. Subfigure (**A**) illustrates the longitudinal changes in body weight across experimental groups throughout the study period. While control animals maintained steady weight progression, diabetic rats exhibited marked weight reduction following alloxan-induced diabetes establishment. Administration of *Lactobacillus rhamnosus GG* alleviated but did not completely prevent the weight loss associated with diabetes. The prophylactic efficacy of probiotic supplementation was evidenced by significantly preserved body mass in treated diabetic animals compared to their untreated counterparts by study termination (** *p* < 0.01, *** *p* < 0.001). Subfigure (**B**) demonstrates the glycemic modulatory effects of *L. rhamnosus GG*, revealing markedly reduced fasting blood glucose (FBG) concentrations in probiotic-treated rats relative to diabetic controls (* *p* < 0.05). This substantial improvement in glycemic homeostasis underscores the metabolic benefits of targeted microbiome intervention.

**Figure 2 biology-14-00723-f002:**
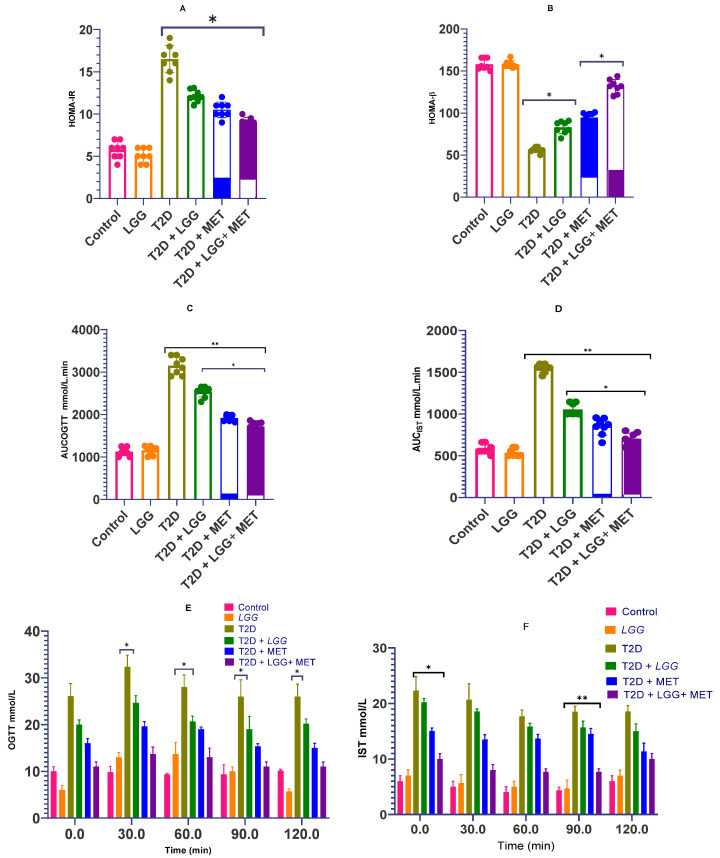
(**A**–**F**) Effects of *Lactobacillus rhamnosus GG* supplementation on glucose homeostasis and insulin dynamics in diabetic rats. This figure summarizes the impact of probiotic and/or metformin treatment on HOMA-IR, HOMA-β, AUC for OGTT and IST, and temporal glucose/insulin response curves at week 8. Data are presented as mean ± SD (*n* = 8–10/group). Statistical significance is indicated as follows: * *p* < 0.05, ** *p* < 0.01, vs. T2D control unless otherwise specified. Where applicable, inter-treatment comparisons are indicated within the figure.

**Figure 3 biology-14-00723-f003:**
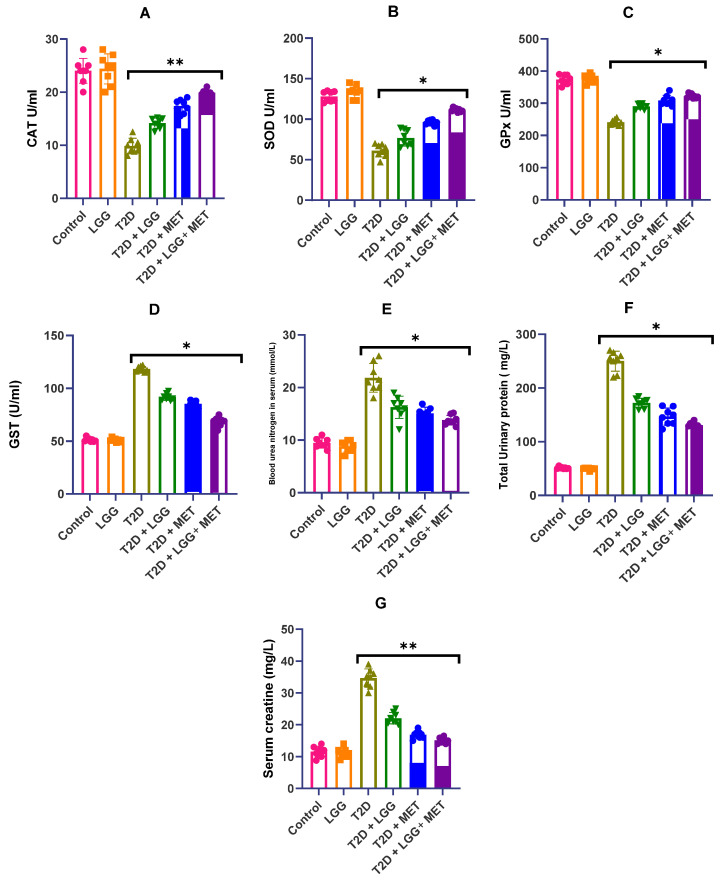
(**A**–**G**) *Lactobacillus rhamnosus GG* was used to study the effects of the bacteria on renal function and antioxidant biomarkers in diabetic rats. This figure depicts the influence that supplementation with *L. rhamnosus* GG, both on its own and in conjunction with metformin, had on the activities of antioxidant enzymes and kidney functional indicators in diabetic rats. Serum creatinine, blood urea nitrogen (BUN), urine albumin excretion, and the enzymatic activities of catalase (CAT), superoxide dismutase (SOD), and glutathione peroxidase (GPx) are among the parameters assessed. The statistics are provided as the mean plus the standard deviation, based on a sample size of eight to ten rats per group. The symbols * *p* < 0.05 and ** *p* < 0.01 are used to denote statistical significance in comparison to the diabetes group. Comparisons with the *L. rhamnosus GG* group are indicated by the symbols.

**Figure 4 biology-14-00723-f004:**
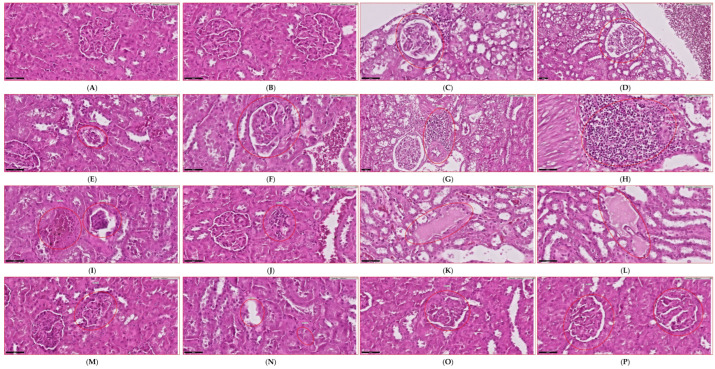
A comprehensive evaluation of renal functional parameters and histological changes following *Lactobacillus rhamnosus GG* intervention in diabetic nephropathy. Representative micrographs (**A**–**P**) and quantitative analyses (**Q**–**S**) demonstrating the nephroprotective effects of *L. rhamnosus GG* in experimental diabetes. Control group (**A**) exhibits normal renal architecture with intact glomeruli and tubular structures. *L. rhamnosus GG*-supplemented control (**B**) shows preserved renal morphology similar to control. Diabetic kidneys (**C**–**F**) display marked pathological alterations including glomerular hypertrophy, inflammatory cell infiltration, tubular damage, and interstitial fibrosis. Diabetic animals receiving metformin treatment (**G**,**H**) show moderate improvement in renal histopathology. Diabetic animals receiving *L. rhamnosus GG* treatment (**I**,**J**) display significant amelioration of histopathological features with preserved glomerular integrity and reduced inflammatory changes. Combined treatment with *L. rhamnosus GG* and metformin (**K**,**L**) demonstrates the most pronounced therapeutic effect with near-normal renal architecture. Higher magnification images (**M**–**P**) reveal ultrastructural improvements in podocyte morphology and basement membrane thickness following single and combined interventions. Quantitative analyses confirm significant reductions in albuminuria (**Q**), glomerular filtration rate abnormalities (**R**), and renal tissue fibrosis (**S**) across treatment groups, with the most substantial improvements observed in the combined therapy group. The therapeutic hierarchy of efficacy is: T2D + *L. rhamnosus GG* + Metformin > T2D + *L. rhamnosus GG* > T2D + Metformin > T2D (untreated). Data are presented as the mean plus or minus the standard error of the mean (SEM), with a total of ten animals belonging to each group. The following equation was used to determine statistical significance: * *p* < 0.05, ** *p* < 0.01, in comparison to the control group; in comparison. In order to conduct the histological examination, panels (**A**–**P**) were stained with hematoxylin and eosin (H&E). On each figure, the original magnification bar is shown.

**Figure 5 biology-14-00723-f005:**
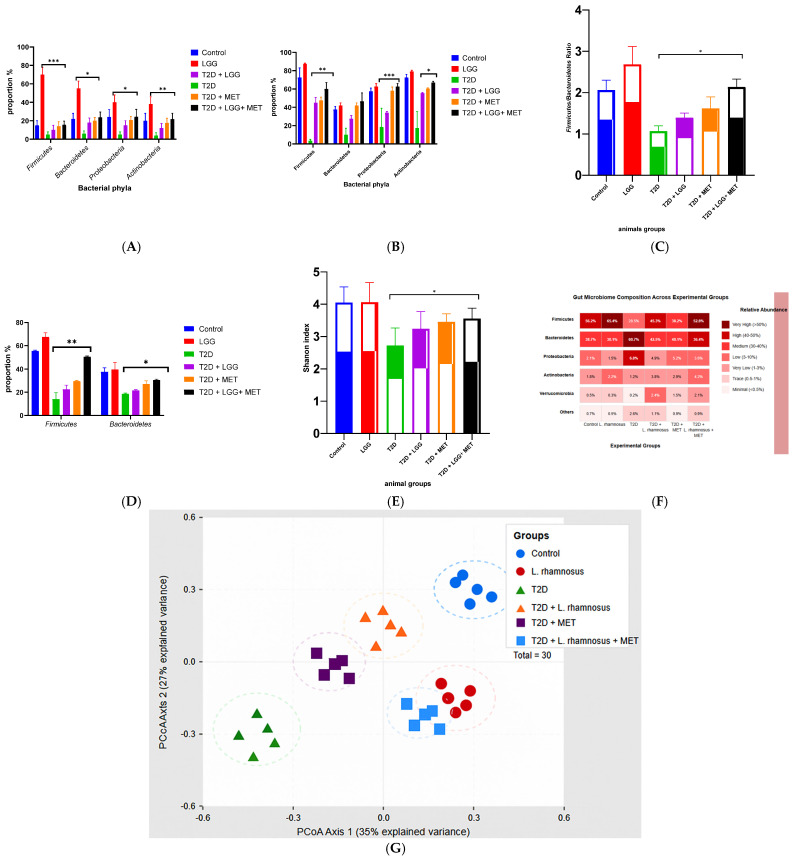
(**A**) Relative Abundance of Bacterial Phyla Across Experimental Groups. Stacked bar chart illustrating phylum-level taxonomic composition. Diabetic rats showed reduced *Firmicutes* (35%) and increased *Bacteroidetes* (35%) compared to controls (*Firmicutes* 75%). *LGG* supplementation partially restored *Firmicutes* abundance to 55%, indicating microbiome reconstitution. (**B**) Shannon Diversity Index of Gut Microbiota. Bar graph showing α-diversity across groups. Diabetic animals exhibited significantly reduced diversity (2.3 ± 0.28) versus controls (4.7 ± 0.12, *p* < 0.001). *LGG*-treated rats displayed significant improvement (4.0 ± 0.16), indicating recovery of microbial richness and evenness. (**C**) *Firmicutes*/*Bacteroidetes* Ratio as a Marker of Microbiome Balance. Bar graph depicting the F/B ratio, which was markedly reduced in diabetic rats (0.7 ± 0.18). *LGG* intervention significantly restored this ratio to 1.7 ± 0.11 (*p* < 0.01), indicating correction of gut dysbiosis. (**D**) Heatmap of Bacterial Phylum Abundance. Heatmap showing relative phylum-level abundances across groups. *LGG* administration resulted in *Firmicutes* recovery (55%) and normalization of minor phyla compared to diabetic rats (*Firmicutes* 35%), supporting microbiota-modulatory effects. (**E**) Principal Coordinate Analysis (PCoA) of Microbial Community Structure. PCoA plot based on weighted UniFrac distances shows distinct clustering of groups. *LGG*-treated microbiota clustered closer to controls, reflecting restored community architecture (explained variance: PC1 = 35%, PC2 = 27%). (**F**) Functional Potential of Gut Microbiota. Comparative bar chart indicating microbial functional profiles. *LGG* supplementation enhanced the expression of pathways involved in SCFA biosynthesis and carbohydrate metabolism, approaching control-like functional potential. (**G**) Compositional Analysis of *Firmicutes* and *Bacteroidetes*. Bar chart showing phylum-level proportions. Diabetic rats showed disrupted *Firmicutes* (35%) and *Bacteroidetes* (35%) balance. *LGG* restored *Firmicutes* to 55% and reduced *Bacteroidetes* to 25%, indicating microbiome modulation. Data are presented as mean ± SEM. Statistical significance was evaluated using one-way ANOVA followed by post hoc Tukey’s test. * *p* < 0.05, ** *p* < 0.01, *** *p* < 0.001 vs. T2D group.

## Data Availability

All datasets generated or analyzed during this study are included in the manuscript.

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
