# Peer review of "Gut Microbiome Engineering for Diabetic Kidney Disease Prevention: A Lactobacillus rhamnosus GG Intervention Study"

_biology, 2025, doi:10.3390/biology14060723_

Round 1
Reviewer 1 Report
Comments and Suggestions for Authors
This study investigated the effects of Lactobacillus rhamnosus GG (LGG) in a streptozotocin-induced diabetic rat model. The investigation is of significant scientific interest and provides valuable insights for microbiome-targeted strategies in diabetic kidney disease. Nevertheless, all the visual representations presented in the article necessitate substantial revision. Each figure needs to be meticulously revised and reordered. Several key issues require attention, as detailed below.
- In the last paragraph of the paper's introduction, it is mentioned that metabolomics analysis was used in this study. However, the results section lacks any figures or tables presenting relevant metabolomics data.
- Regarding the figures, Figures 1, 2, and 3 display bar charts for the T2D + LGG, T2D + MET, and T2D + LGG + MET groups, which appear nearly identical. This raises doubts about the accuracy of the data.
- In Result 3.1, the author states in the text that diabetic rats supplemented with LGG exhibited significantly preserved body mass compared to their non-treated diabetic counterparts. However, the results in Figure 1A do not seem to demonstrate that LGG significantly alleviates weight loss in diabetic mice. Please indicate which statistical test was used in Figure 1 and other figures. Please add significance difference markers in Figure 1A and add some explanations to clarify the results more clearly.
- Figure 2 has multiple issues. The sub-figures vary in size, and the legend does not clearly define the experimental results each sub-figure represents. Moreover, the asterisk markers may be mislabeled, and the figure does not indicate which groups show a significant difference with three stars. Although the authors state that the combined intervention group (T2D + LGG + MET) had the most significant decrease in HOMA-IR values, Figure 2A suggests no difference among the T2D + LGG + MET, T2D + LGG, and T2D + MET groups. Similar clarification is needed for Figure 2B-F, as the figures do not reflect the claimed superiority of the combined treatment.​
- Figures 4 and 5 are absent from the text, necessitating a rearrangement of the figures.
- To improve readability, figures should be presented on single pages, avoiding page breaks, as seen in Figures 3, 6, and 7.
- In Figure 6, the icons of some sub - figures are at the top, while those of some sub - figures are at the bottom. Other figures in the text also have such issues. Please unify them.
- Many figures require adjustment to standardize the size of its sub-figures.​
- Page 4, the second paragraph of “Biochemical and Metabolic Assessments” lacks an indentation at the beginning.
- Page 6, in the “Alpha Diversity Analysis” section, the font size of “According to” does not match the rest.
- Additionally, Lactobacillus rhamnosus, L. rhamnosus, and LGG are not consistently italicized in the text and figures, which should be corrected for academic consistency.
Author Response
Response to Reviewer
I sincerely thank the reviewer for taking the time to provide thoughtful and constructive feedback on my manuscript entitled:
"Gut Microbiome Engineering for Diabetic Kidney Disease Prevention: A Lactobacillus rhamnosus GG Intervention Study."
I have carefully addressed each of the reviewer’s comments, and corresponding revisions have been implemented throughout the manuscript. Below is a point-by-point response to the reviewer’s observations.
Major Concerns
- "In the last paragraph of the paper's introduction, it is mentioned that metabolomics analysis was used in this study. However, the results section lacks any figures or tables presenting relevant metabolomics data."
Response:
Thank you for this valuable observation. The mention of “metabolomics analysis” in the original version was indeed imprecise. This study did not involve direct metabolomic profiling. Instead, I employed predictive functional profiling using PICRUSt2, which infers microbial metabolic potential from 16S rRNA gene data. These results are thoroughly reported in Section 3.6.6.
To resolve this inconsistency, I have revised the final paragraph of the Introduction. The updated version now accurately reflects the methodology, replacing “metabolomics analysis” with:
“predictive functional analysis (via PICRUSt2), and detailed histopathological evaluations…”
This correction aligns the Introduction with the actual methods and results of the study and avoids reference to any unreported analytical technique.
- "Figures 1, 2, and 3 display bar charts for the T2D + LGG, T2D + MET, and T2D + LGG + MET groups, which appear nearly identical. This raises doubts about the accuracy of the data."
Response:
I appreciate the reviewer’s close inspection of the data presentation. After thoroughly rechecking the underlying data, I confirm its validity. However, I acknowledge that the initial visual representation lacked clarity due to scale compression.
Therefore, I have:
- Replotted Figures 1–3 with improved scaling, clearer color contrast, and enhanced error bars.
- Included statistical significance markers to highlight intergroup differences.
- "In Result 3.1, the author states that LGG significantly preserved body mass; however, Figure 1A does not show this clearly."
Response:
Thank you for this constructive comment. Upon review, I acknowledge that the original wording may have overstated the effect of LGG on body weight preservation. To enhance clarity and accuracy:
- I have revised the statement in Section 3.1 to:
"Administration of Lactobacillus rhamnosus GG alleviated, but did not completely prevent, the weight loss associated with diabetes."
- Additionally, I have updated Figure 1A to include statistical significance markers (e.g., *p < 0.01), clearly indicating intergroup differences.
- The statistical method applied—One-way ANOVA with Tukey’s post hoc test—has also been specified in both the figure legend and the Methods section to provide full transparency.
This correction ensures that the text now more accurately reflects the observed data and its biological implications.
- "Figure 2 has multiple issues regarding sub-figure size, unclear legends, mislabeled asterisks, and unsubstantiated claims of combined therapy superiority."
Response:
Thank you for this detailed and valuable observation. I have carefully revised both the visual presentation and the corresponding descriptive text in response to the noted issues:
Sub-figure standardization: All sub-figures in Figure 2 have been resized and aligned to maintain consistent layout and proportion across the panel.
Legend clarity: The figure legend has been rewritten to clearly describe the content and comparisons of each sub-panel. Statistical annotations (p values and group comparisons) have been explicitly defined to prevent misinterpretation.
Statistical markers correction: All asterisk markers were re-checked and corrected in accordance with the actual statistical analyses conducted using one-way ANOVA with Tukey’s post hoc test.
Revision of textual claims (Section 3.2): The original overstatement regarding the superiority of the combined LGG + MET therapy has been moderated. The revised text now indicates that while the combined treatment group showed numerical improvements in HOMA-IR, HOMA-β, and AUC values, statistical superiority over monotherapies was not consistently observed across all measured parameters. This adjustment ensures the narrative remains scientifically accurate and aligned with the actual results.
These revisions collectively improve the clarity, accuracy, and integrity of the manuscript’s presentation. I sincerely appreciate the reviewer’s critical input, which helped enhance the overall quality of this section.
- "Figures 4 and 5 are absent from the text."
Response:
Thank you for highlighting this issue. Upon thorough review, it was determined that the confusion stemmed from a numbering error rather than missing content. The manuscript originally referenced seven figures, but the actual number of figures included is five.
To resolve this, I have:
Corrected the figure numbering throughout the manuscript to accurately reflect the total number of figures.
Ensured that all five figures are properly cited, described, and integrated within the Results section in their correct sequence.
This correction ensures consistency between figure references and the visual content provided. I sincerely thank the reviewer for bringing this matter to attention.
- "Figures should be on single pages, avoiding breaks (Figures 3, 6, and 7)."
Response:
I agree with this suggestion. I have reformatted all figures to appear on dedicated pages, avoiding page breaks, and ensuring enhanced readability and continuity.
- "Figure 6 icons are inconsistently positioned; sub-figure layouts should be unified."
Response:
Corrected. I have standardized the placement of all sub-figure labels (e.g., A–P) and ensured consistent formatting across all multi-panel figures.
- "Many figures require adjustment to standardize sub-figure sizes."
Response:
I have re-rendered all relevant figures with uniform panel dimensions and resolution. The formatting now adheres to MDPI’s publication standards.
Minor Concerns
- "Page 4 – the second paragraph of 'Biochemical and Metabolic Assessments' lacks indentation."
Response:
This has been corrected. The formatting now follows standard paragraph indentation.
- "Page 6 – the font size of 'According to' in the Alpha Diversity section is inconsistent."
Response:
Corrected. The font size has been standardized throughout the manuscript for consistency.
- "Lactobacillus rhamnosus, L. rhamnosus, and LGG are not consistently italicized."
Response:
I conducted a detailed formatting review, and all scientific names have now been italicized consistently in both the main text and figure labels in accordance with journal guidelines.
Closing Statement
Once again, I express my sincere gratitude to the reviewer for the insightful and constructive feedback, which significantly contributed to improving the overall quality, clarity, and scientific value of this manuscript. I hope that the revised version now meets the standards for publication in Biology.
Respectfully,
The Author
Reviewer 2 Report
Comments and Suggestions for Authors
This manuscript evaluated the protective effects of the probiotic Lactobacillus GG against diabetic kidney injury through in vivo intervention experiments in rats, exploring the potential mechanisms by which Lactobacillus GG modulates the gut-kidney axis. Multi-omics analysis revealed that the probiotic improved gut microbiota balance (Prevotella/Bacteroides ratio) and metabolic dysfunction in diabetic rats. The findings demonstrate certain clinical relevance, suggesting that probiotic supplementation may serve as a potential adjunctive therapy. However, the article contains several notable issues that require revision to strengthen its conclusions.
Question 1
Generally speaking, streptozotocin (STZ) injection model mimics type 1 diabetes, whereas this study focuses on renal injury in type 2 diabetes (T2D), raising concerns about the appropriateness of the chosen animal model. The authors should justify why STZ was selected over high-fat diet/obesity models (e.g., db/db mice) for investigating T2D-associated diabetic kidney disease (DKD).
Question 2
The statistical significance annotations in every statistical chart in the manuscript are problematic. Additionally, the figure numbering is discontinuous. The OGTT and ITT data in Figure 2 could be presented as line graphs, which would allow for more intuitive visualization of blood glucose changes. Figure 3 is directly followed by Figure 6. In Section 3.6, all figure citations lack corresponding numerical labels.
Question 3
The Discussion section states that Lactobacillus rhamnosus GG administration demonstrates significant renoprotective effects through multiple mechanisms including microbiome remodeling, metabolic improvement, and inflammation modulation. However, the Results section lacks any data regarding inflammatory factor detection or metabolite analysis. Additional experiments addressing these aspects should be incorporated to support the claimed mechanisms.
Question 4
The figure legend for Figure 7 contains mismatched panel descriptions and omits several images. Please carefully verify and revise.
Author Response
Response to Reviewer
I would like to sincerely thank the reviewer for their thoughtful comments, valuable insights, and constructive critique of my manuscript entitled:
"Gut Microbiome Engineering for Diabetic Kidney Disease Prevention: A Lactobacillus rhamnosus GG Intervention Study."
Below is my detailed point-by-point response to the reviewer’s comments:
Comment 1:
The streptozotocin (STZ) model mimics type 1 diabetes, whereas this study focuses on type 2 diabetic kidney disease. Please justify the use of this model rather than obesity or HFD-based models (e.g., db/db mice).
Response:
I appreciate this important observation. In this study, I used a combination model involving high-fat diet (HFD) feeding followed by low-dose STZ injection, which is a well-established approach to simulate type 2 diabetes mellitus (T2DM) and its associated complications, including diabetic kidney disease (DKD). This method induces insulin resistance followed by partial beta-cell dysfunction, reflecting the pathophysiological features of T2DM more accurately than STZ alone. This clarification has been added to the Materials and Methods section, with appropriate references cited.
Comment 2:
Statistical annotation symbols in all figures are inconsistent; figures are not clearly ordered. Suggest using line graphs for OGTT/ITT. Also, Figure 3 precedes Figure 6, and section 3.6 lacks proper figure references.
Response:
Thank you for pointing this out. I have made the following adjustments:
- Unified all statistical annotations across the figures to ensure consistency.
- Replaced the bar charts for OGTT and IST with line graphs, which better represent temporal glucose changes.
- Renumbered and reordered the figures to maintain logical flow (now totaling five figures), correcting the issue where Figure 3 preceded Figure 6.
- Updated all figure citations in Section 3.6 to ensure they are accurate and correspond to the correct figures.
Comment 3:
The discussion refers to anti-inflammatory and metabolic effects of L. rhamnosus GG, but no cytokine or metabolite measurements were provided.
Response:
I appreciate the reviewer’s valuable comment. Indeed, the current study did not include direct measurements of inflammatory cytokines or metabolomic profiling. The anti-inflammatory and metabolic interpretations were instead derived from predictive functional metagenomic analysis using PICRUSt2, which infers microbial community functions based on 16S rRNA sequencing data, and from well-characterized taxonomic shifts—such as the enrichment of SCFA-producing genera and suppression of pro-inflammatory taxa.
These inferences are supported by literature and are frequently employed in microbiome research where direct metabolite data are not available. I have now revised the Discussion section to clearly specify that these conclusions are based on predictive analysis rather than direct measurements, and I have explicitly stated this as a limitation of the study. Given the focus of this work on microbial composition and functional prediction, rather than host molecular responses, I opted not to include additional cytokine or biochemical analyses at this stage. However, I fully recognize the importance of such data and intend to include targeted biochemical and inflammatory profiling in future studies to further substantiate the mechanistic insights.
Comment 4:
Figure 7 legend is inconsistent with figure panels and omits descriptions for some images. Please revise.
Response:
Thank you for identifying this oversight. Upon review, I found inconsistencies in figure numbering which caused confusion. In the revised version of the manuscript, there are now only five figures in total. The misnumbering that suggested seven figures has been corrected. The figure legend in question (now Figure 5) has been thoroughly revised to match all sub-panels, and clear, concise descriptions have been added to align the visual data with the accompanying narrative.
In conclusion, I am deeply grateful to the reviewer for their constructive comments, which have greatly improved the clarity, structure, and scientific accuracy of the manuscript. I believe the revisions now address all points raised and strengthen the overall quality of the submission.
Sincerely,
The Author
Reviewer 3 Report
Comments and Suggestions for Authors
Well-presented and enjoyable study. Two small suggestions:
- I would spend a few more words in the introduction in the dedication of the Firmicutes/Bacteroidetes ratio and its implications,
- It would be helpful to express, perhaps even in with a schematic table, the outcomes of the 6 groups
Author Response
Response to Reviewer
I would like to sincerely thank the reviewer for the positive and thoughtful evaluation of my manuscript. I greatly appreciate the constructive feedback, which has helped improve the clarity and depth of the work.
Comment 1: "I would spend a few more words in the introduction in the dedication of the Firmicutes/Bacteroidetes ratio and its implications."
Response:
Thank you for this valuable suggestion. I have revised the final part of the Introduction to include a clearer explanation of the Firmicutes/Bacteroidetes (F/B) ratio, emphasizing its relevance as a key microbial biomarker for gut dysbiosis, metabolic inflammation, and disease progression in diabetic conditions. This addition enhances the contextual background for its role in the present study.
Comment 2: "It would be helpful to express, perhaps even in with a schematic table, the outcomes of the 6 groups."
Response:
As recommended, I have included a supplementary schematic table (Supplementary Table S1) that summarizes the major outcomes across all six experimental groups. This table provides a concise comparative overview of the physiological, histological, and microbiome-related findings, which improves readability and facilitates interpretation.
Once again, I thank the reviewer for their encouraging remarks and insightful feedback, which contributed meaningfully to the refinement of the manuscript.
Sincerely,
The Author
Round 2
Reviewer 1 Report
Comments and Suggestions for Authors
After revision, the article has significantly improved. However, a few minor issues still remain.
- Figure 2, Figure 3, and Figure 5 (A-E) exhibit misaligned sub-images, and the text within these images shows significant variations in size and font. Kindly adjust them for enhanced visual appeal.
- Figure 4 and Figure 5 continue to have page break problems. Please ensure they are typeset on a single page.
Author Response
Response to Reviewer’s Comments
I would like to sincerely thank the esteemed reviewer for the constructive feedback and for recognizing the improvements made in the revised manuscript. I have carefully addressed the remaining points as outlined below:
Reviewer Comment:
Figure 2, Figure 3, and Figure 5 (A–E) exhibit misaligned sub-images, and the text within these images shows significant variations in size and font. Kindly adjust them for enhanced visual appeal.
Response:
Thank you for this important remark. I have revised Figures 2, 3, and 5 to ensure that all sub-panels are properly aligned. I have also unified the font type and size across all labels within these figures to enhance their visual consistency. The updated figures have been incorporated into the revised manuscript accordingly.
Reviewer Comment:
Figure 4 and Figure 5 continue to have page break problems. Please ensure they are typeset on a single page.
Response:
I appreciate this observation. I have reformatted Figures 4 and 5 to prevent any page breaks. Each figure is now properly typeset on a single page to ensure optimal readability and presentation quality.
I hope these final revisions meet the reviewer’s expectations, and I remain grateful for the helpful insights provided.